# Social media-based Health Education plus Exercise Program (SHEEP) to improve muscle function among young-old adults with possible sarcopenia in the community: A feasibility study protocol

**Ya Shi** [1,2] *, **Emma Stanmore** [1,3,4], **Lisa McGarrigle** [1,3], **Chris Todd** [1,3,4]

**1** Faculty of Biology, School of Health Sciences, Medicine & Health, University of Manchester, Manchester, United Kingdom, **2** School of Nursing & School of Public Health, Yangzhou University, Yangzhou, Jiangsu province, China, **3** School of Health Sciences and Manchester Academic Health Science Centre (MAHSC), University of Manchester, Manchester, United Kingdom, **4** Manchester University NHS Foundation Trust, Manchester, United Kingdom

\* ya.shi@postgrad.manchester.ac.uk

**Data Availability Statement:** Deidentified research data will be made publicly available when the study is completed and published.

## Abstract

### Background

Despite the comparatively high prevalence of possible sarcopenia among young-old adults in the community, there is currently no available and effective social media-based intervention to increase the awareness and change the behavior of the target population to prevent sarcopenia. Using co-design methodology, we developed a multicomponent intervention strategy of health education and exercise for sarcopenia prevention utilizing the TikTok platform.

### Objectives

The primary purpose of this study is to examine the feasibility and acceptability of the social media-based intervention to enhance muscle function in community-dwelling young-old adults with possible sarcopenia.

### Methods

This protocol outlines the entire research procedure for a prospective single-arm pre-post feasibility study employing a mixed-method design, which will be conducted between May 2024 and September 2024. Thirty-five older adults aged 60–69 years with possible sarcopenia will be recruited from two communities in Changsha, China. Using the TikTok platform, participants will be required to view a total of seven health education videos in the first week, and each video lasts four to six minutes. Then, participants will receive six-week multi-component exercise through TikTok, with at least three sessions/week, 30 minutes/session, and moderate intensity. Data collection will be conducted in baseline, week 1, 4, 7, 10 and 13. The primary outcomes will include evaluating recruitment capability, data collection

**Funding:** YS is funded by the University of Manchester - China Scholarship Council Joint Scholarship (202108320049), and CT is funded by the National Institute for Health and Care Research Senior Investigator Award (NIHR200299). LM is funded through the National Institute for Health and Care Research (NIHR). She was funded by NIHR Senior Investigator Award to Prof Todd (NIHR200299) until March 2023 and then by NIHR Policy Research Unit in Older People and Frailty (PR-PRU-1217-2150). As of 01.01.24, the unit has been renamed to the NIHR Policy Research Unit in Healthy Ageing (NIHR206119). The views expressed are those of the author(s) and not necessarily those of the NIHR or the Department of Health and Social Care or its partner organisations. No funding bodies had any role in study design, data collection and analysis, decision to publish, or preparation of the manuscript.

**Competing interests:** The authors have declared that no competing interests exist.

procedure, outcome measurement, intervention procedures' acceptability, researchers' ability to manage and implement the study, among others. The secondary outcome is to compare standard measures for muscle function (e.g. handgrip strength, skeletal muscle mass, physical performance), body composition (e.g. body fat, body mass index, bone mineral), and other measures (e.g. perceived knowledge, personal motivation, behavioral skills). Finally, all participants will be offered a semi-structured interview to assess their in-depth experiences with the intervention and research process.

## Discussion

This study will be the first social-media based multicomponent intervention program for community young-old adults with possible sarcopenia to improve their muscle function, awareness and behavior of preventing sarcopenia. Findings will generate new evidence regarding the use of social media in health education for improving awareness of sarcopenia prevention, as well as the feasibility of using social media to influence participants' behavioral changes through exercise. This may help researchers identify ways to optimize acceptability and efficacy of the SHEEP intervention for the targeted population.

## Trial registration

ISRCTN registry, ISRCTN17269170, Registered 14 September 2023.

## Introduction

Possible or probable sarcopenia is defined as age-related low muscle strength, with or without reduced physical performance [1,2]. It is highly prevalent among older adults in the community according to ample evidence [3–7]. Several factors such as increased age [5–9], male gender [5,10], living in rural areas [7], having other disease conditions (e.g., diabetes, hypertension, chronic lung diseases, chronic kidney disease, heart disease, arthritis, osteoarthritis, osteoporosis, rheumatism, frailty, malnutrition, depression and psychiatric conditions) [4–7,9–11], physical inactivity [4–6,8,9], and having a history of falls [7] were associated with higher prevalence of possible sarcopenia. Moreover, receiving a higher level of education is associated with a lower risk of possible sarcopenia [6]. It is notable that some factors associated with this condition are preventable, treatable, and modifiable. Hence, the authoritative international associations for sarcopenia prevention recommend possible sarcopenia as a significant threshold in medical practice to trigger assessment of causes and initiate intervention [1,2].

There are a limited number of interventions specifically designed to prevent possible sarcopenia in community-dwelling young-old adults. Our previous scoping review indicated that most studies of sarcopenia interventions recruited participants older than 70 years, while young-old adults aged 60–69 years received less attention [12]. For example, Merchant et al. [13] discovered that community-based dual-task exercise could improve handgrip strength, physical performance, perceived health, cognition, and reduce falls and frailty in Singaporean older adults (mean age 75.9y) with possible sarcopenia. Jones et al. [14] conducted a one-group, pre-post experimental study to examine the Healthy Beat Acupunch (non-invasive punching movements based on traditional Chinese medicine) exercise program on sarcopenia prevention and found that it was simple, safe, appropriate, and beneficial for practice by Australian older adults (mean age 81.2y) with probable sarcopenia and could improve participants'

gait speed. Another recent Taiwanese study examined the impacts of vitality acupunch exercise on older adults (mean age 80.5y) with possible sarcopenia, concluding that acupunch exercise has positive effects on the muscle mass, hand grip strength, and sleep quality of participants after a six-month intervention [15]. Furthermore, Pan et al. [16] designed a multicomponent intervention strategy consisting of nutrition, resistance exercise, and health education for com- munity- dwelling Chinese seniors (mean age 71.6y) with possible sarcopenia, which increased their muscle mass and handgrip strength and decreased the duration of the timed up-and-go test and the five-times chair stand test. Additionally, the proportion of non-pharmaceutical interventions for older individuals with possible sarcopenia (11.9%) was substantially lower than that for sarcopenia (72.9%) in the community [12].

Meanwhile, our recently published scoping review revealed a paucity of research exploring the intervention type of 'health education plus exercise' in sarcopenia prevention [12]. While exercise is a widely recognized non-pharmacological strategy for managing sarcopenia [1,2], the effectiveness of health education remains ambiguous due to its relatively limited focus in this domain [3,12]. Our scoping review showed that the total number of health education interventions (15.5%) is small when compared with exercise (52.8%) or nutrition interventions (34.5%) to prevent possible sarcopenia or sarcopenia in the community [12]. Our review also indicated that more than 80% of the educational materials promoted in health education inter- ventions were not explicitly related to sarcopenia [12]. This may be the primary cause for the unsatisfactory outcomes of current health education for sarcopenia prevention [3,12]. In addi- tion, the identified intervention methods for delivering health education to prevent possible sarcopenia or sarcopenia were traditional, including group-based classes, face-to-face interac- tions and printed leaflets/materials [12].

Social media has become a promising platform for health education and behavior change in prevention of diseases in recent years, but there are no reports in the sarcopenic area. For instance, a recent study demonstrated that participation in a Twitter social media support group for breast cancer patients (of which 32.9% were between 55~74 years) increased overall breast cancer knowledge and decreased anxiety [17]. Another study determined that a video clip followed by a text post was the most effective method of informing hypertensive individu- als about the primary prevention of cardiovascular diseases using the Instagram social media platform [18]. Facebook advertising has also been shown to be effective for disseminating hypertension knowledge to older adults in research; with the click-through rates for older (65 years or older) Chinese Americans higher than those for younger (45-54 years), at 15% versus 4% respectively [19]. In addition to knowledge dissemination, social media-based interven- tions are widely used in the promotion of positive behavior change. A national web-based cross-sectional survey in China reported that social media, including TikTok, WeChat and Weibo, was an effective tool to disseminate pandemic news and disease knowledge, and was also beneficial for promoting behaviors to prevent COVID-19 among the public [20]. A sys- tematic review focusing on behavior change interventions through social media found that a majority of studies reported positive outcomes and most of the studies targeted either physical activity or a combination of behaviors that included physical activity (e.g., physical activity, diet and nutrition, and weight loss or weight maintenance) [21]. Moreover, a rapid evidence- based review revealed that studies using social media in interventions have mostly showed superior results to more traditional methods and suggest that public health institutions, clini- cians, and other stakeholders consider the use of social media in their interventions targeting people affected with chronic disease [22].

Therefore, our research team co-created a Social-media based Health Education plus Exer- cise Program (SHEEP) for preventing possible sarcopenia through behavior change in com- munity-dwelling young-old adults. Full details of the development study are published

elsewhere [3,23]. In brief, we constructed a SHEEP Conceptual Model primarily guided by the Behavior Change Wheel (BCW) [24] and the conceptual model for Lifestyle-integrated Functional Exercise using smartphones and smartwatches (eLiFE) [25], with the objective of developing intervention content and the monitoring process. Then, we co-designed a multicomponent social media-based (TikTok) intervention based on the SHEEP Conceptual Model. After a three-week preliminary test, the results from semi-structured interviews indicated that the participants complied well with the SHEEP intervention and provided positive feedback for both health education and exercise components. Not only did participants subjectively report that their grip and leg strength increased after exercise, they perceived that their mood, sleep, and diet also improved. Simultaneously, participants reported that the program positively altered their attitudes and behaviors toward sarcopenia prevention, and that they would continue to exercise after the study was completed. They were also willing to recommend the program to others. To ensure a more impartial assessment of the potential impact of SHEEP intervention on the intended population, we devised this extended feasibility study prior to proceeding with a prospective RCT.

## Study aims and objectives

The main aim of this study is to assess the feasibility and acceptability of the SHEEP intervention to improve muscle function among young-old adults with possible sarcopenia in the community. According to the feasibility checklist created by Orsmond and Cohn [26], the specific objectives of this research are: 1) to assess recruitment capability and sample characteristics; 2) to assess data collection procedures and outcome measurements; 3) to assess intervention and study procedures' acceptability and suitability; 4) to assess resources and the ability to manage and implement the study and intervention; and 5) to assess participant responsiveness to intervention.

## Materials and methods

### Study design

A single-arm prospective pre-post study will be conducted to evaluate the feasibility, acceptability, and preliminary impact of the SHEEP on the prevention of possible sarcopenia in community-dwelling young-old adults. In this seven-week intervention with six-week follow-up study, both quantitative and qualitative methods will be employed to evaluate the outcomes. This research protocol was based on the Standard Protocol Items: Recommendations for Interventional Trials—13 (SPIRIT-13) checklists (S1 and S2 Files) and Template for Intervention Description and Replication (TIDieR) checklist (S3 File).

### Study setting

This feasibility study will be conducted in Changsha, Hunan Province, China, with the assistance of our Chinese collaborators. The recruitment and intervention procedure will be carried out in two communities, including Sanchaji Community Health Service Centre and Guanshaling Community Health Service Centre.

### Eligibility criteria

Different scholars have different interpretations of the definition of young-old, like 60–69 or 65–74 [27,28]. Due to a lack of research in the 60–69 year age group (as evidenced from our scoping review [12]) and considering this feasibility study will be carried out in China where ages 60+ are considered an older person [29–31], 60–69 years old is defined as young-old in

this study. Trained health professionals and the principal investigator (PI) will approach older adults to take part in this study and they will use the inclusion and exclusion criteria to ensure that only eligible people take part. The following inclusion and exclusion criteria will be considered for the recruitment of participants:

· *Inclusion criteria.*

1. 60~69 years old

2. Chinese residents living in the community

3. Have a smart phone with an internet connection at home, existing users of TikTok or have a willingness to download and use TikTok

Individuals with sarcopenia screening questionnaire: Five-item (strength, walking assistance, rising from a chair, stair-climbing, and falls) combined with calf circumference [SARC-CalF] ≥11 [2,32]

1. Individuals with possible sarcopenia, as defined by low Grip Strength [M:<28 kg, F:<18kg] in accordance with the 2019 Asian working group consensus on diagnosis criteria for sarcopenia [2]

2. Be able to attend the community health center (at their own expense if travel is required) on six occasions for the duration of the study

3. Informed consent for screening and research

· *Exclusion criteria.*

1. Unable to communicate or independently complete learning and exercise using TikTok

2. Serious or unstable medical illness, such as severe cardiovascular or respiratory conditions, mental disorder, dementia, etc.

3. Currently undertaking regular exercise (e.g. Tai Chi and Square Dance) and exercise intensity is moderate (≥ 150mins/week)

4. Contraindications for the use of Bioelectrical Impedance Analysis (BIA), such as people with implanted cardiac pacemakers, or other electronic devices or metal grafts, or with significant pitting oedema, or with limb dysfunction or body paralysis, or while taking medications that affect body composition, such as diuretics or glucocorticoids.

## Recruitment, consent and withdrawal

The PI will visit the two community health centers to present this research to managers and get the approval of the managers. Then, the managers will assist the PI to distribute paper information leaflets to older people who visit community health centers and send electronic information leaflets to community WeChat groups. If older adults are interested in this study after reviewing the leaflet, they can contact the PI through the managers or directly via the contact information listed on the leaflet. To reduce the risk of participants coming to meet with the PI unnecessarily, the PI will conduct a preliminary eligibility screening over phone/email (e.g. confirm items 1–3 and 6 on eligibility list, and exclusion criteria). If confirmed, the PI will invite the participants to meet with them in an office provided by the community health center to tell them more about the study and determine whether they are eligible based on items 4 and 5 (SARC-CalF and grip strength). Potential participants who meet the eligibility

requirements will be provided with a full explanation of research that informs them of the detailed research content, the confidentiality of personal information, the potential benefits, the potential risks, and a coping strategy for the risks, should they wish to know. A Participant Information Sheet will be provided for subjects who are still interested and willing to participate in the study, and time will be allowed for consideration (at least 24 hours). Finally, those who agree to participate in the study will be required to provide (written or verbal) informed consent (face-to-face or through phone calls) prior to data collection. If participants choose verbal consent, the PI will read each statement of the consent form and request their response via a dedicated mobile phone. Verbal consent will be separately recorded and stored.

## Research ethics

This study has been approved by the University of Manchester Research Ethics Committee (Project ID: 2024-19302-34066), and permissions have already been granted by collaborators in the Community Nursing Department of Xiang Ya Nursing School, Central South University, China. Amendments to the protocol, study design or supporting documents will be approved by the University of Manchester Research Ethics Committee prior to implementation. All documents will be maintained with version number and tracked changes.

## Sample size

One arm of 35 eligible participants will be recruited. A sample size of 30 participants after attrition (attrition rate 10%~15%) is normally used for a feasibility study [33,34]. Convenience sampling will be utilized in this research. The target population in the two communities will be divided into four categories based on age (60–65 and 66–69 years) and gender (male and female), to determine if there are differences in the acceptability of the intervention process among subgroups of individuals.

## Randomization, allocation, and blinding

This study is a one-arm pre-post feasibility study, so randomization and allocation are not required currently. A cluster RCT is planned, so there is no need to evaluate the feasibility of individual randomization. Blinding is not appropriate for this study as there is only one arm, and the intervention contents include health education, which explicitly specifies that this is a course aimed at the prevention of sarcopenia. Therefore, it can be readily distinguished that the exercise strategy in our study is for muscle health and sarcopenia prevention by researchers and participants when the intervention begins.

## Intervention

Participants will receive the SHEEP intervention strategy [23], which consists of one-week of health education and six-weeks of exercise. Participants will be required to view a total of seven health education videos posted on TikTok in the first week, including the following: 1) What is sarcopenia? 2) What are the common influencing factors in sarcopenia? 3) What are the adverse effects of sarcopenia? 4) What are the manifestations of sarcopenia in daily life? 5) What are the common screening methods for sarcopenia? 6) What are the exercise methods to prevent sarcopenia? 7) What are the nutritional methods to improve sarcopenia? Each video lasts four to six minutes. There is no restriction on the number of views, so participants are allowed to determine the number of videos they watch per day.

Participants will attend a six-week exercise training course on TikTok following completion of the health education learning. The duration of the exercise is fixed at 30 minutes and

comprises four types of exercise: 3-minute warm-up training, 8-minute aerobic training, 16-minute resistance training, and 3-minute flexibility training. The multicomponent exercise training is informed by the evidence-based FaME exercise program (https://laterlifetraining. co.uk/real-life-implementation-of-fame-does-it-work/), OTAGO exercise program (https:// www.livestronger.org.nz/assets/Uploads/acc116 2-otago-exercise-manual.pdf), and our previous study on intervention development which involved collaboration from stakeholders (physicians, nurses, physiotherapists, older participants) [23]. The training is delivered through pre-recorded videos on TikTok. The frequency of exercise must be at least 3 times/week, and participants can increase the frequency of training per week based on their own preferences. The overall exercise intensity is moderate. The resistance exercise is performed using two 500ml or 1000ml or 1500ml or 2000ml capacity water-bottles. Participants can start from 500ml or 1000ml, then adding 100-500ml of weight every three weeks. The increase in weight is not fixed and can be adjusted according to preference. When the body feels no burden after completing an exercise session, for instance the body does not heat, sweat or muscles do not appear slightly sore after exercise, we suggest appropriately increasing the weight. These water bottles are all made of plastic. 500ml and 1000ml water bottles are more common in mineral water bottles or juice bottles, while 1500ml and 2000ml water bottles are more common in sports water bottles.

Finally, participants will be followed up weekly over another 6 weeks of exercise. The researcher will help them record their exercise diary and feeling through a short telephone call or text message each week.

## Outcomes

As this is a small feasibility study, the main aim is to assess the feasibility and acceptability of the entire research process, from recruitment to the delivery of the intervention and the use of the outcome measurements. The secondary objectives are to compare standard measures at baseline, seven-week post-intervention, and six-week follow-up for muscle-related outcomes, other body measurements as detailed below, and study adherence to evaluate the potential efficacy of the intervention. The outcome measures will be collected at different time points to identify an appropriate primary outcome and estimate parameters for a sample size calculation for an RCT in the future. Final semi-structured interviews with all participants will be conducted to investigate the viability of the intervention and study procedures in greater detail.

### · *Primary aim*

Evaluate the feasibility and acceptability of the research procedures, including:

a. Assessing recruitment capability and participant characteristics: Recording the recruitment process, including participant numbers screened, eligible, approached, consented, and excluded after screening; then calculating the relevant rates.

b. Assessing intervention and study procedures' acceptability and suitability: a) Participants' evaluation of the intervention and study procedures; b) Whether the rationale for participant dropout is associated with the intervention and study procedures; c) Participants adherence to the intervention including the compliance rate for completion of the entire study, health education, and exercise, respectively; d) Follow-up rate = Number of participants who complete the follow-up stage / Number of participants who complete the six-week exercise intervention.

c.  Assessing data collection procedures and outcome measurements: a) Time needed to collect data; b) Experiences of the secondary outcome measures, such as whether they feel fatigued or burdened.

d.  Assessing resources and the ability to conduct the study and intervention: a) Duration of time required to complete each step of this study; b) Ability to complete the study within the allotted time; c) Capacity of researchers to collaborate with community healthcare professionals; d) Evaluate researchers' data organization, management, and analysis ability.

e.  Assessing participant responsiveness to intervention: a) Investigate subjective experiences of the participants during the intervention; b) Measure objective outcomes before and after intervention.

· *Secondary outcomes*

Table 1 outlines the two types of secondary outcomes that will be evaluated at various points throughout the study. The first category consists of muscle-related outcomes, including muscle strength, muscle mass, and physical performance. The second category includes additional body parameters such as anthropometric indices, nutrition state, perceived knowledge, personal motivation, behavioral skills, and monitoring of behavior change. The questionnaire survey, like MNA-SF, SEMCD-6, and the Self-Management for Chronic Disease Scale, will be conducted after the physical measurements are assessed. The PI will ask participants the questionnaire questions verbally and will record their answers electronically. If participants wish to

**Table 1. Overview of instruments and time of outcome measures.**

| MEASURES | INSTRUMENTS | $T_0$ | Intervention | | | Follow-up | |
|---|---|---|---|---|---|---|---|
| | | | $T_1$ | $T_4$ | $T_7$ | $T_{10}$ | $T_{13}$ |
| Social-demographic information | Demographics questionnaire (Self-developed) | X | | | | | |
| **Primary outcome** | | | | | | | |
| Feasibility and acceptability | Research record sheet (Self-developed) | X | X | X | X | X | X |
| **Secondary outcome** | | | | | | | |
| Handgrip strength | Digital hand-held dynamometer | X | | X | X | X | X |
| Anthropometric measurements (e.g. weight, height) | Stadiometer + Scale + Plastic tape | X | | X | X | X | X |
| Body composition (e.g. muscle mass, body fat) | Bioelectrical impedance analyzer device | X | | X | X | X | X |
| Physical performance | 4-Meter Walk Test | X | | X | X | X | X |
| | Five Times Sit to Stand Test | X | | X | X | X | X |
| Nutrition state | Mini-Nutritional Assessment Short Form | X | | X | X | X | X |
| Perceived knowledge | 21 true or false quizzes related to sarcopenia (Self-developed) | X | X | X | X | X | X |
| Personal motivation | Self-efficacy for Managing Chronic Disease 6-item Scale | X | | X | X | X | X |
| Behavioral skills | Self-management Behavior for Chronic Disease Scale | X | | X | X | X | X |
| Behavior change monitoring | Exercise diary (Self-developed) | X | X | X | X | X | X |
| | Exercise Adherence Rating Scale (EARS) | X | | X | X | X | X |
| | Records of sharing sarcopenia-related information (Self-developed) | X | X | X | X | X | X |
| | Willingness to formulate habits of regular exercise after study (Self-developed) | | | | | | X |
| | Records of exposure percentage of exercise and sarcopenia related videos (Self-developed) | X | X | X | X | X | X |

Note: $T_0$ = baseline; $T_1$ = week 1; $T_4$ = week 4; $T_7$ = week 7; $T_{10}$ = week 10; $T_{13}$ = week 13.

see the questionnaires, they will be provided with a paper copy. Those measurements will take about 30–40 minutes in total to complete, which will be assessed in an office provided by the community health center.

### · *Handgrip strength*

Handgrip strength (kg) will be measured with a digital hand-held dynamometer (EH101, Xiangshan Inc, Guangdong, China). In a seated position with the elbow flexed at 90 degrees, the shoulder attached to the thorax, and the wrist in a neutral position (0 degrees), the maximal grip strength is measured. The measurements are taken three times on each side (left and right), with a 30-second interval between each assessment [35].

### · *Anthropometric measurements*

Height will be measured to the nearest 0.1 cm using a stadiometer, and weight will be measured to the nearest 0.1 kg using a scale. Calf circumference and abdominal circumference will also be measured using a tape measure for both legs and the navel, respectively, to the nearest 0.1 cm.

### · *Body composition*

Body composition measurement will be conducted using a multi-frequency bioelectrical impedance analyzer device (InBody S10; Biospace, Seoul, Korea) according to the manufacturer's guidelines. BIA calculates body composition by comparing the conductivity of various tissues based on their distinct biological properties. As a cell approaches an ideal spherical shape, its conductivity decreases, as it is proportional to water content and, more specifically, electrolytes. Since adipose tissue is composed of round-shaped cells and contains relatively little water in comparison to other tissues such as muscle, the conductivity of the body will decrease as body fat increases. In order to conduct a multi-segment frequency analysis, electrodes are set at six precise tactile points of the body. Using six different frequencies (1 kHz, 5 kHz, 50 kHz, 250 kHz, 500 kHz, 1000 kHz), more than 30 impedance measurements are obtained from the right and left arms, trunk, and right and left legs [36]. The following body index parameters will be chosen for analysis in this study: skeletal muscle mass (kg), skeletal mass index (kg/m$^2$), upper-extremity skeletal muscle mass (kg), lower-extremity skeletal muscle mass (kg), trunk skeletal muscle mass (kg), body fat (kg), body fat percentage (%), body mass index (kg/m$^2$), protein (kg), and bone mineral (kg).

### · *Physical performance*

Physical performance will be measured by using a 4-Meter Walk Test (4- MWT, m/s) and a Five Times Sit to Stand Test (FTSST, s).

First, the participants' self-selected walking speed will be measured using a 4-MWT (with 2m for acceleration/deceleration) [37]. At the beginning and end of the timed walkway, space will be provided for participants to accelerate and decelerate outside of the data collection area to reduce gait variability introduced during these phases. Participants will be instructed to "walk at a comfortable, normal pace" until reaching the end of the marked path; and each participant must accomplish three consecutive measurements. A researcher will measure walking time using a stopwatch (ZSD-013, China), starting it when the participant's lead leg crosses the first marker and stopping it when it crosses the second marker. To prevent interobserver variation, all stopwatch measures will be conducted by the same researcher. Throughout the testing session, participants will be provided with breaks as required.

Second, participants will sit on an armless chair with a 43 cm (cm) seat height to commence the FTSST [38]. Each participant will be instructed to sit with their back against the chair's backrest and their arms crossed over their torso. The researcher will then demonstrate the proper technique for performing the test, including coming to a full stand, which is defined as an upright trunk with extended hips and knees. A researcher will measure the time using a stopwatch (ZSD-013, China). Timing will commence when the researcher says "go" and end when the participant's glutes touch the seat after the fifth stand [39]. The participants must stand and sit five times "as quickly as possible" and without physical assistance. The researcher will not use encouraging words or body language to encourage participants to speed up their performance.

### · *Nutritional status*

The Mini-Nutritional Assessment Short Form (MNA-SF), a valid nutritional screening instrument applicable to geriatric health care professionals, will be used to evaluate nutritional status [40]. The MNA-SF consists of six items and one anthropometric measurement (either body mass index [BMI] or calf circumference [CC]) to identify individuals at risk of malnutrition. Maximum MNA-SF score is 14, with scores≤11 indicating possible malnutrition and scores ≥12 confirming normal nutritional status [41]. MNA-SF, having fewer items than Full-MNA, requires less measuring time while achieving similar predictive power as Full-MNA [40,42].

### · *Perceived knowledge*

Participants' perceived knowledge of sarcopenia prevention will be assessed by using true or false questions. The 21 questions reflect the main points of the seven topics in the sarcopenia prevention health education strategy. Final evaluation of the participants' mastery of health education knowledge will be based on the proportion of correct responses. These sarcopenia quizzes were developed in our previous study [23], which involved a literature search, followed by two rounds of PPI consultations (eight older adults, four sarcopenia and health education experts) and two rounds of focus group interviews (31 older adults with possible sarcopenia). All these stakeholders reviewed and approved the content of the questions which lends itself to content validity.

### · *Personal motivation*

Self-Efficacy for Managing Chronic Disease 6-item Scale (SEMCD-6) developed by the Stanford Patient Education Research Centre in a Chronic Disease Self-Management Program will be used to identify and measure the confidence level of older adults in adopting appropriate behavioral skills to prevent and manage chronic disease [43,44]. Scores on this scale are calculated as the means of the six items, including self-efficacy to manage symptoms (4 items) and self-efficacy to manage disease (2 items), each measured on Likert scales ranging from 1 (not at all confident) to 10 (totally confident). The higher the score, the greater the participant's confidence in completing the disease management mission. Measures of SEMCD-6 reliability have been found to be high (= 0.91), and test-retest reliability has been reported to be statistically significant (r = 0.87).

### · *Behavioral skills*

Self-management Behavior for Chronic Disease Scale also developed by the Stanford Patient Education Research Centre will be used to identify and measure older adults' actual behavioral skills in daily life for preventing and managing chronic disease [43–45]. The scale includes

exercise management (5 items), cognitive symptom management (6 items) and communication with physicians (3 items). Exercise management is measured by examining the total time spent engaging in physical activities, such as stretching and strengthening exercises, walking, swimming, bicycling, and aerobics (categorized respectively as 0: none, 1:<30min/ week, 2: 30-60min/week, 3: 1-3h, or 4: >3h/week). Cognitive symptom management is measured on a 6-point Likert scale, ranging from never to always, and includes distraction, progressive muscle relaxation, imagery, positive thinking, and self- distance. Communication with physicians is measured as the number of times the participants prepared a list, asked questions, and discussed problems with the provider on a 6-point Likert scale ranging from never to always.

### · *Behavior change monitoring*

Three aspects of behavior change will be monitored throughout the entire research procedure, including: 1) Exercise adherence during the intervention and follow-up periods will be calculated by completion rate of minimum exercise required in the study. The Exercise Adherence Rating Scale (EARS) is a validated 16-question instrument with a 6-question subscale assessing adherence; the remaining questions measure reasons for adherence/non-adherence. As stated in the study [46], EARS is a valid and reliable tool that allows for the assessment of compliance with suggested home exercise. The six items comprise a unidimensional scale that exhibited favorable measurement features, including satisfactory internal consistency and strong test-retest reliability. 2) Percentage of participants who actively view and share sarcopenia-related information to others. 3) Percentage of participants willing to formulate habits of regular exercise to prevent sarcopenia at the end of the current study. 4) Exposure percentage of exercise and sarcopenia-related videos = Number of exercise and sarcopenia related videos viewed / Twenty videos. TikTok can push videos to users that they are interested in based on big data algorithms; therefore, if participants begin to view more videos about exercises and sarcopenia after health education in our research, TikTok will capture these changes in their viewing habits and push more videos about exercises and sarcopenia to them. Then, researchers can record the number of exercise and sarcopenia-related videos when every twenty videos are viewed on participants' TikTok and observe the exposure percentage of exercise and sarcopenia-related videos to determine if participants' video-scrolling behavior changes.

### · *Social-demographic information*

The participants will respond to single-item questions regarding their age in years, gender, long-term condition, medication, level of education, and any significant health issues that may impact their ability to engage in physical activity, as well as whether they use a mobility aid.

### · *Qualitative Interview*

After the six-week follow-up period, all participants, including those who withdraw at any point, will be offered a semi-structured interview in an office provided by the community health center to assess their experiences with the intervention and research process. This will take about 30min. Family members/ caregivers may also be invited to the interview at the request of the participant.

## Timeline

This will be a 17-week study, including 2-week recruitment, 1-week health education, 6-week exercise intervention, 6-week follow up, and 2-week interview. The expected completion date for the research is between May 2024 and September 2024. The recruitment phase will last

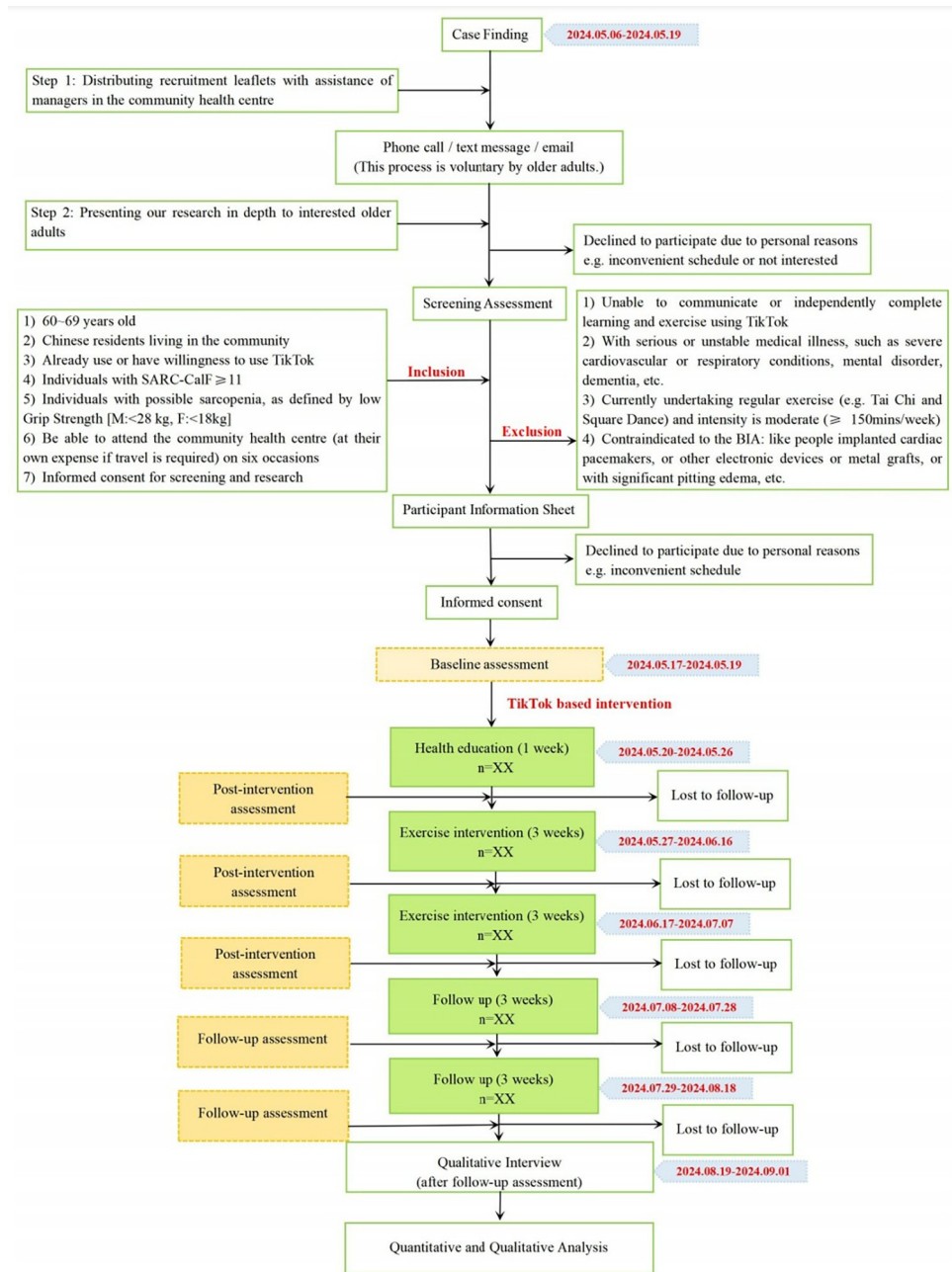

A brief diagram of the feasibility study procedure and timeline

**Fig 1. A brief diagram of the feasibility study procedure and timeline.**

approximately ten days because, during the intervention development phase, our research group collaborated with medical personnel at community health service centers and gained experience in recruitment. Fig 1 illustrates the comprehensive research procedure and precise timetable.

## Data collection and analysis

The PI, who is qualified and experienced with all of the measurements, will be responsible for data collection. To promote the collection of high-quality data, the PI will double-check the collected data.

As for quantitative data analysis, descriptive statistics will be utilized to summarize the percentage of recruitment, baseline participant characteristics (e.g. age, gender, and educational background), attrition rate, compliance rate, height, weight, etc. 95% confidence intervals (CIs) of mean values will be used to determine the difference of secondary outcomes before and after intervention, such as handgrip strength, skeletal muscle mass, body fat, etc. All statistical analyses will be conducted using SPSS Statistics 27.0 (IBM Corp., Armonk, NY, USA). P Values $<0.05$ will be interpreted as indicating significance.

Regarding the collection of qualitative data, all face-to-face semi-structured interviews after the follow-up period will be audio recorded using an encrypted digital recorder. If participants prefer a telephone interview, the conversation will be recorded on an encrypted Dictaphone. All encrypted recordings will be uploaded to a secure server at the University of Manchester and then deleted from the recording device. Moreover, each interview will last between 30 minutes and one hour, and participants will only be asked to participate in one. During the data analysis stage, we will analyze the qualitative data using thematic analysis. Qualitative data will be transcribed verbatim and potentially identifying information will be removed. The transcripts will be compared to the original recordings, corrected as needed, and anonymized. To become familiar with the qualitative data, researchers will first read all transcripts. Themes will be developed based on the interview guide and research team discussion to identify the key characteristics of the qualitative data. Then, codes will be generated for each line of each transcript and categorized accordingly. Before finalizing the major themes, each category must be re-evaluated and scrutinized, and groups of major categories will be refined. Two researchers will perform and review the coding process to ensure a double check, followed by a discussion to ensure the validity of the data. To increase the transparency of the interpretation, the Chinese-to-English and English-to-Chinese translation of quotations will be performed in this study. QSR International's NVivo 12 qualitative analysis software will be used to assist and facilitate the coding and analysis process. We will refer to the theoretical standpoints from the SHEEP Conceptual Model [23], the BCW [24] and the conceptual model for eLiFE [25] when analyzing the qualitative data in this study.

## Data management

The PI will preserve the confidentiality of participants taking part in the study and fulfil transparency requirements under the General Data Protection Regulation for health and care research. Data and all appropriate documentation will be confidentially and securely stored for a minimum of 10 years after the completion of the study, including the follow-up period.

## Study monitoring

The PI will oversee participant recruitment, intervention and follow-up. In addition, one community leader in each cooperative community health center will assume overall responsibility for participant identification and recruitment. One team comprised of research professionals from the United Kingdom and China will be responsible for ensuring the overall quality of research data. All principal researchers will hold a monthly online meeting to report on the project's progress and discuss its problems and solutions. A thorough risk assessment has been conducted, and potential patient, organizational, and study hazards, as well as their likelihood of occurrence and potential consequences, have been considered.

## Discussion

This research aims to test the feasibility and acceptability of a social-media based multicomponent intervention (health education + exercise) for community young-old adults with possible sarcopenia. Utilizing a mixed-methods approach will enable the investigation of barriers and facilitators to a successful implementation of the intervention, in order to adequately prepare for the future implementation of a randomized controlled trial. Additionally, based on quantitative data analysis, this study will be the first to observe the potential impact of a social-media intervention on the awareness and behavior change of older individuals in preventing sarcopenia, with the ultimate goal of enhancing their muscle performance.

However, there are still some limitations in this research. First of all, the feasibility study was developed as a single-arm pre-post study rather than a feasibility randomized controlled trial due to limited resources (few researchers, limited time and funding). In the future, a multi-center feasibility randomized controlled trial will be considered to explore the feasibility of the research and validate potential effects between different intervention groups, including social media-based health education plus exercise, social media-based health education, social media-based exercise, and control group. Secondly, this study will recruit samples from only two communities, potentially leading to a sample that is not adequately representative. Enhancing the representativeness of future studies can be achieved by expanding the recruitment area. Thirdly, the exercise design failed to incorporate older people who depend on mobility aids such as wheelchairs or crutches. Future studies could explore the creation of customized exercise programs for this specific demographic. Fourthly, this study will exclude older people who do not have Internet access. Therefore, exploring the possibility of integrating both online and offline interventions to prevent sarcopenia is also an innovation area in the future.

The research data and findings of this study will be presented in peer-reviewed journals and scientific conferences, as well as be shared through a series of public engagement events including webinars. The anonymized participant level dataset will not be publicly available but will be available from the PI upon reasonable request.

## Supporting information

**S1 File. SPIRIT 2013 checklist.**
(PDF)

**S2 File. Example template of recommended content for the schedule of enrolment, interventions, and assessments.**
(PDF)

**S3 File. TIDieR checklist.**
(PDF)

## Author Contributions

**Conceptualization:** Ya Shi, Emma Stanmore, Lisa McGarrigle, Chris Todd.

**Data curation:** Ya Shi.

**Formal analysis:** Ya Shi.

**Funding acquisition:** Ya Shi, Lisa McGarrigle, Chris Todd.

**Investigation:** Ya Shi.

**Methodology:** Ya Shi, Emma Stanmore, Lisa McGarrigle, Chris Todd.

**Project administration:** Ya Shi, Emma Stanmore, Lisa McGarrigle, Chris Todd.

**Resources:** Ya Shi, Emma Stanmore, Lisa McGarrigle, Chris Todd.

**Software:** Ya Shi.

**Supervision:** Emma Stanmore, Lisa McGarrigle, Chris Todd.

**Validation:** Ya Shi, Emma Stanmore, Lisa McGarrigle, Chris Todd.

**Visualization:** Ya Shi.

**Writing – original draft:** Ya Shi.

**Writing – review & editing:** Emma Stanmore, Lisa McGarrigle, Chris Todd.

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
