## [Decision Letter · Decision Letter 0]

18 Nov 2024

PONE-D-24-15585Social-media based health education plus exercise programme (SHEEP) to improve muscle function among young-old adults with possible sarcopenia in the community: A feasibility study protocolPLOS ONE

Dear Dr. SHI,

Thank you for submitting your manuscript to PLOS ONE. After careful consideration, we feel that it has merit but does not fully meet PLOS ONE’s publication criteria as it currently stands. Therefore, we invite you to submit a revised version of the manuscript that addresses the points raised during the review process.

We look forward to receiving your revised manuscript.

Kind regards,

Marina De Rui, MD PhD

Academic Editor

PLOS ONE

Reviewers' comments:

Reviewer's Responses to Questions

**Comments to the Author**

1. Does the manuscript provide a valid rationale for the proposed study, with clearly identified and justified research questions?

Reviewer #1: Partly

Reviewer #2: Yes

2. Is the protocol technically sound and planned in a manner that will lead to a meaningful outcome and allow testing the stated hypotheses?

Reviewer #1: Partly

Reviewer #2: Yes

3. Is the methodology feasible and described in sufficient detail to allow the work to be replicable?

Reviewer #1: No

Reviewer #2: Yes

4. Have the authors described where all data underlying the findings will be made available when the study is complete?

Reviewer #1: No

Reviewer #2: Yes

5. Is the manuscript presented in an intelligible fashion and written in standard English?

Reviewer #1: No

Reviewer #2: Yes

6. Review Comments to the Author

You may also provide optional suggestions and comments to authors that they might find helpful in planning their study.

Reviewer #1: Dear author,

Thank you for submitting your manuscript “Social-media based health education plus exercise programme (SHEEP) to improve muscle function among young-old adults with possible sarcopenia in the community: A feasibility study protocol”, in which you explain how you want to examine the feasibility and acceptability of the social

media-based intervention to enhance muscle function in community-dwelling young-old

adults with possible sarcopenia.

I have read your manuscript with great interest, but advised the editor to reject it. The brief reason is as follows: I find your study relevant, but the study protocol not clear and structured enough. I had a hard time reading through the manuscript and distilling what you are going to do in your study and what has already been done in your scoping review.

I suggest you rewrite the manuscript and submit it to another journal.

Some more in-depth comments on the text:

• Have your manuscript checked by a native English speaker – the text doesn’t read well - check the proper use of definite and indefinite articles (the and a), this is mixed up in the text and confusing to read.

• Be consistent in the use of ‘possible sarcopenia’ and ‘probable sarcopenia’, it’s now used both, which is confusing.

• The introduction could be a lot more concise – the paragraph on prevalence of sarcopenia, risk factors and modifiable risk factors could be shortened to a few sentences

• Explain earlier what you mean by ‘young old adults’ – this is not a common term and it became only clear in the methods (not mentioned in abstract or introduction)

• Nice acronym, SHEEP, but be more clear where it stands for: Social-media based Health Education plus Exercise Programme (SHEEP) ?

• It’s not clear if you are focusing on health education or sarcopenia prevention.

• Add line numbers to the text, as requested in the format of the journal

• The recruitment section is too long, too extensive, it gets lost in the weeds – be more concrete and concise how people were recruited.

• Of course with a one-arm pre-post feasibility study blinding is not necessary, part of the procedures etc but I don’t get your explanation of why blinding in this study is not appropriate.

• It’s nutritional status, not nutritional state.

• I don’t understand what you mean by “MNA-SF appears to combine the predictive power of Full-MNA with an assessment period that is sufficiently brief”

• Figure 1: it’s not clear what is inclusion and what is exclusion (it’s the arrows, but not clear to the reader)

good luck with your intervention!

Reviewer #2: The authors conduct a prospective single-arm study to evaluate the feasibility and acceptability of the social media-based intervention to improve muscle function among young-old adults with possible sarcopenia. They will recruit 35 participants for this study. In general, the study is well designed and clearly presented.

1. This study will be a one-arm study. What is the reasoning behind this design not to recruit the control group?

2. It isn’t clear how authors will determine the feasibility and acceptability in this study.

7. PLOS authors have the option to publish the peer review history of their article (what does this mean?). If published, this will include your full peer review and any attached files.

Reviewer #1: No

Reviewer #2: No

---

## [Author Response · Author response to Decision Letter 0]

11 Dec 2024

Response to comments

Dear Editor and Reviewers: 

Manuscript ID number: PONE-D-24-15585

Title: Social-media based health education plus exercise program (SHEEP) to improve muscle function among young-old adults with possible sarcopenia in the community: A feasibility study protocol

Thank you so much for your insightful feedback on our manuscript. We have revised and responded to each comment in accordance with your suggestion, as demonstrated below.

Journal requirements

We have modified our manuscript according to the requirements of two documents, PLOSOne_formatting_sample_main_body 

and 

PLOSOne_formatting_sample_title_authors_ affiliations.

Thank you for this comment. We have reviewed and checked the reference list to ensure that it is complete and correct. (P17-20)

Reviewers’ comments

1. Does the manuscript provide a valid rationale for the proposed study, with clearly identified and justified research questions?

Reviewer #1: Partly

Reviewer #2: Yes

According to the suggestions of reviewer 1, we have simplified the research background and supplemented key evidence to make the research rationale and research questions clearer. (P3-5)

2. Is the protocol technically sound and planned in a manner that will lead to a meaningful outcome and allow testing the stated hypotheses?

Reviewer #1: Partly

Reviewer #2: Yes

We have detailed descriptions of research methods and checked to prevent ambiguity in the protocol. (P6-17)

3. Is the methodology feasible and described in sufficient detail to allow the work to be replicable?

Reviewer #1: No

Reviewer #2: Yes

According to the suggestions of reviewer 1 and reviewer 2, we have added more details in the methods section to allow the work to be replicable. (P6-17)

4. Have the authors described where all data underlying the findings will be made available when the study is complete?

Reviewer #1: No

Reviewer #2: Yes

We have added this description at the end of the discussion section.

See lines 615-619

The research data and findings of this study will be presented in peer-reviewed journals and scientific conferences, as well as be shared through a series of public engagement events including webinars.

The anonymized participant level dataset will not be publicly available but will be available from the principal investigator upon reasonable request. (P17)

This is as per the approved data management plan reviewed by the University of Manchester ethics committee.

5. Is the manuscript presented in an intelligible fashion and written in standard English?

Reviewer #1: No

Reviewer #2: Yes

The manuscript has been refined thoroughly by two of our native English-speaking research members, including typographical or grammatical errors.

6.Review Comments to the Author

Reviewer #1: 

Dear author,

Thank you for submitting your manuscript “Social-media based health education plus exercise programme (SHEEP) to improve muscle function among young-old adults with possible sarcopenia in the community: A feasibility study protocol”, in which you explain how you want to examine the feasibility and acceptability of the social media-based intervention to enhance muscle function in community-dwelling young-old adults with possible sarcopenia.

I have read your manuscript with great interest, but advised the editor to reject it. The brief reason is as follows: I find your study relevant, but the study protocol not clear and structured enough. I had a hard time reading through the manuscript and distilling what you are going to do in your study and what has already been done in your scoping review.

We thank reviewer 2 for their positive comments about the paper. We have made minor corrections as requested by the editor and trust the paper is now ready for publication. 

Currently, our research is rather novel in sarcopenia area. Our most recent intervention development study protocol about the SHEEP was also published in the journal of PLOS ONE, which has received citations and endorsements from other scholars in sarcopenia area. It indicates that our series of studies on SHEEP merits consideration. This feasibility study is a progression and continuation of our intervention development study. This feasibility study protocol has been reviewed and directed by ethical reviewers during the ethical review phase, so ensuring clarity in the study process to some extent. We have endeavoured to amend and enhance it in accordance with your feedback. We remain optimistic about publishing this protocol in PLOS ONE, as it is presently the most suitable journal for our submission.

Some more in-depth comments on the text:

• 1. Have your manuscript checked by a native English speaker – the text doesn’t read well - check the proper use of definite and indefinite articles (the and a), this is mixed up in the text and confusing to read.

Thank you so much for your recommendations, and two of our native English-speaking research members have amended the protocol thoroughly.

• 2. Be consistent in the use of ‘possible sarcopenia’ and ‘probable sarcopenia’, it’s now used both, which is confusing.

Thanks for the comment. Possible sarcopenia and probable sarcopenia refer to the same condition. ‘Possible sarcopenia’ is referenced in Asian Working Group for Sarcopenia 2019, while ‘probable sarcopenia’ is cited in European Working Group on Sarcopenia in Older People 2018. We are therefore following the nomenclature of the paper cited – so if a paper used the phrase “possible sarcopenia” we use that phrase, and if the paper used the phrase “probable sarcopenia” we use that phrase. However, it is clear these are actually synonyms. 

• 3. The introduction could be a lot more concise – the paragraph on prevalence of sarcopenia, risk factors and modifiable risk factors could be shortened to a few sentences

Thank you so much for your advice. We have condensed these parts (prevalence of sarcopenia, risk factors and modifiable risk factors) into a few sentences. (P3)

• 4. Explain earlier what you mean by ‘young old adults’ – this is not a common term and it became only clear in the methods (not mentioned in abstract or introduction)

Thank you so much for your suggestion. We have added the explanation of ‘young old adults’ in both the introduction and methods section. (P3, P6)

• 5. Nice acronym, SHEEP, but be more clear where it stands for: Social-media based Health Education plus Exercise Programme (SHEEP) ?

We appreciate your acknowledgement. 

We have modified accordingly in the title and introduction section. (P1, P5)

• 6. It’s not clear if you are focusing on health education or sarcopenia prevention.

Thank you very much for your comments. We have revised some of the descriptions in the introduction section to focus on both health education and exercise. Our inclusion criteria were older adults with possible sarcopenia in the community, while possible sarcopenia is the initial threshold of screening and intervention phase recommended for sarcopenia prevention by both Asian Working Group for Sarcopenia 2019 and European Working Group on Sarcopenia in Older People 2018. Hence, we focused on health education plus exercise intervention strategy for sarcopenia prevention. 

We do not understand why this reviewer is struggling here. We have looked at the manuscript again and believe it is absolutely clear that the health education component alongside the exercise component comprise a sarcopenia prevention strategy. (P4, P5)

• 7. Add line numbers to the text, as requested in the format of the journal

Thanks for reminding us. We had added line numbers to the text.

• 8. The recruitment section is too long, too extensive, it gets lost in the weeds – be more concrete and concise how people were recruited.

Thank you so much for your suggestion. We have refined the recruitment section's description for enhanced clarity. (P7)

• 9. Of course with a one-arm pre-post feasibility study blinding is not necessary, part of the procedures etc but I don’t get your explanation of why blinding in this study is not appropriate.

Thank you for this comment. 

Perhaps the reviewer would like to consider different levels of blinding in their comment.

We included this information about the reason why single blinding of the participants was not appropriate as the health education component explicitly specifies that this is a course aimed at the prevention of sarcopenia. Likewise, as this is a single arm study blinding of researcher collecting data (double blinding) or analyst (triple blinding) are also not appropriate.

See lines 285-289

Blinding is not appropriate for this study as there is only one arm, and the intervention contents include health education, which explicitly specifies that this is a course aimed at the prevention of sarcopenia. Therefore, it can be readily distinguished that the exercise strategy in our study is for muscle health and sarcopenia prevention by researchers and participants when the intervention begins. (P8)

• 10. It’s nutritional status, not nutritional state.

Thank you for your correction. We have modified it accordingly. (P12)

• 11. I don’t understand what you mean by “MNA-SF appears to combine the predictive power of Full-MNA with an assessment period that is sufficiently brief”

Thanks for your question. MNA-SF, having fewer items than Full-MNA, requires less measuring time while achieving similar predictive power as Full-MNA. We have revised this sentence in the relevant section. (P12)

• 12. Figure 1: it’s not clear what is inclusion and what is exclusion (it’s the arrows, but not clear to the reader)

Thank you so much for your advice. We have made the necessary changes in Figure 1 to make both inclusions and exclusions clear. (P14-15)

Reviewer #2: 

The authors conduct a prospective single-arm study to evaluate the feasibility and acceptability of the social media-based intervention to improve muscle function among young-old adults with possible sarcopenia. They will recruit 35 participants for this study. In general, the study is well designed and clearly presented.

Thank you very much for your recognition and patient review of our manuscript.

• 1. This study will be a one-arm study. What is the reasoning behind this design not to recruit the control group?

Thanks. This is a good question, as we had also considered this question when we originally designed the study. We have already discussed this in the research limitation section.

The feasibility study was developed as a single-arm pre-post study rather than a feasibility randomized controlled trial due to limited resources (few researchers, limited time and funding). In the future, a multi-center feasibility randomized controlled trial will be considered to explore the feasibility of the research and validate potential effects between different intervention groups, including social media-based health education plus exercise, social media-based health education, social media-based exercise, and control group. (P16-17)

• 2. It isn’t clear how authors will determine the feasibility and acceptability in this study.

Thank you for this comment. We have clarified in the outcomes section about how we are evaluating the feasibility and acceptability of the research from five main aspects. (P9-10)

Finally, we really appreciate all the time and effort you put into this document, as well as all your insightful suggestions for the minor revisions of this manuscript to make it more readable and clearer. Thank you so much once more!

Best wishes

---

## [Editor Report · Decision Letter 1]

17 Dec 2024

Social media-based Health Education plus Exercise Program (SHEEP) to improve muscle function among young-old adults with possible sarcopenia in the community: A feasibility study protocol

PONE-D-24-15585R1

Dear Dr. SHI,

We’re pleased to inform you that your manuscript has been judged scientifically suitable for publication and will be formally accepted for publication once it meets all outstanding technical requirements.

Kind regards,

Marina De Rui, MD PhD

Academic Editor

PLOS ONE
---

## [Editor Report · Acceptance letter]

22 Dec 2024

PONE-D-24-15585R1 

PLOS ONE

Dear Dr. Shi, 

I'm pleased to inform you that your manuscript has been deemed suitable for publication in PLOS ONE. Congratulations! Your manuscript is now being handed over to our production team.

Kind regards, 

on behalf of

Dr. Marina De Rui 

Academic Editor

PLOS ONE